# Adverse Childhood Events Significantly Impact Depression and Mental Distress in Adults with a History of Cancer

**DOI:** 10.3390/cancers16193290

**Published:** 2024-09-27

**Authors:** Oluwole A. Babatunde, Katherine Gonzalez, Nosayaba Osazuwa-Peters, Swann Arp Adams, Chanita Hughes Halbert, Frank Clark, Anusuiya Nagar, Jessica Obeysekare, Eric Adjei Boakye

**Affiliations:** 1Department of Psychiatry, Prisma Health, Greer, SC 29650, USA; frank.clark@prismahealth.org (F.C.); anu.nagar@prismahealth.org (A.N.); jessica.obeysekare@prismahealth.org (J.O.); 2School of Medicine, California University of Science and Medicine, Colton, CA 92324, USA; katherine.gonzalez@duke.edu; 3Department of Head and Neck Surgery & Communication Sciences, Duke University School of Medicine, Durham, NC 27710, USA; nosa.peters@duke.edu; 4Department of Population Health Sciences, Duke University School of Medicine, Durham, NC 27710, USA; 5Duke Cancer Institute, Durham, NC 27701, USA; 6Department of Epidemiology and Biostatistics, Arnold School of Public Health, University of South Carolina, 915 Greene Street, Columbia, SC 29208, USA; adamss@mailbox.sc.edu; 7Biobehavioral Health and Nursing Science Department, College of Nursing, University of South Carolina, 1601 Greene Street, Columbia, SC 29208, USA; 8Department of Population and Public Health Sciences, University of Southern California, Los Angeles, CA 90032, USA; hughesha@usc.edu; 9Norris Comprehensive Cancer Center, University of Southern California, Los Angeles, CA 90089, USA; 10School of Medicine-Greenville, University of South Carolina, Greenville, SC 29605, USA; 11Department of Public Health Sciences, Henry Ford Health System, Detroit, MI 48202, USA; eadjei1@hfhs.org; 12Department of Otolaryngology—Head and Neck Surgery, Henry Ford Health System, Detroit, MI 48202, USA; 13Henry Ford Health + Michigan State University Health Sciences, Detroit, MI 48202, USA; 14Department of Epidemiology and Biostatistics, Michigan State University College of Human Medicine, East Lansing, MI 48824, USA

**Keywords:** adverse childhood experiences, depression, mental distress, cancer survivors, sexual abuse, mental abuse, physical abuse, BRFSS

## Abstract

**Simple Summary:**

This research was inspired by a desire to understand how adverse childhood experiences (ACEs) impact depression among cancer survivors. The study’s findings revealed that cancer survivors with three or more ACEs had a depression rate of 40.8%, compared to 18.7% among those with one to two ACEs and 10.9% among those with no ACEs. Survivors with three or more ACEs were significantly more likely to report experiencing depression. Among the three ACE subtypes—sexual abuse, physical abuse, and having family members with mental illnesses—cancer survivors with family members suffering from mental illnesses had the highest odds of depression, followed by those who experienced physical abuse. These findings provide a crucial basis for health providers to screen cancer survivors for ACEs and depression, guiding proactive screening and treatment of depression in this vulnerable population.

**Abstract:**

**Objectives:** Adverse childhood experiences (ACEs) are linked to a heightened risk of depression. We explored the relationship between ACEs and both depression and mental distress among cancer survivors. **Methods:** This was a cross-sectional analysis using the 2022 Behavioral Risk Factor Surveillance System database of cancer survivors aged ≥18 (n = 14,132). The primary outcome was self-reported history of depression, and the secondary outcome was mental distress. The exposure variable was the number of ACEs, classified as 0, 1–2, and ≥3. Weighted multivariable logistic regression models assessed the association between the number of ACEs and depression and mental distress while adjusting for covariates. **Results:** Approximately 22% of respondents reported experiencing ≥3 ACEs. The prevalence of depression was 21.8%, and mental distress was 15.4%. Compared with cancer survivors who had experienced 0 ACEs, those who had experienced ≥3 (aOR = 3.94; 95% CI, 3.04–5.10) or 1–2 (aOR = 1.85; 95% CI, 1.47–2.32) ACEs had a higher likelihood of reporting depression. Compared with cancer survivors who had experienced 0 ACEs, those who had experienced ≥3 (aOR = 0.67; 95% CI, 0.48–0.93) had a lower likelihood of reporting mental distress. **Conclusions:** This study highlights the impact of ACEs on depression in adulthood among cancer survivors.

## 1. Introduction

The prevalence of mental health disorders is on the rise, posing a growing concern for public health and impacting both illness rates and mortality [1]. In the United States, an estimated 57 million adults are affected by various forms of mental health disorders or illnesses [2]. This figure has more than doubled in the past decade, a trend further exacerbated by the COVID-19 pandemic [3,4]. Consequently, there has been a significant surge in demand for mental health care services in recent years without enough providers to meet this demand [5]. However, the treatment and management of mental illnesses in the United States carry substantial economic implications, with direct healthcare costs estimated at over USD 230 billion [6]. In addition to these direct costs, there are indirect economic repercussions such as decreased productivity, lost wages, and disability [6,7]. The economic burden of addressing mental health disorders is anticipated to continue rising in the coming years, prompting ongoing efforts to mitigate these costs through prevention and innovative treatment approaches [8].

A growing body of research has uncovered a correlation between stressful childhood experiences and the onset of various mental health disorders, particularly depression and anxiety [9]. Adverse childhood experiences (ACEs) refer to psychosocial factors that detrimentally affect the well-being and growth of children and adolescents [10]. These factors encompass emotional and physical abuse, neglect, violence, and other significant challenges encountered by children within and outside the home [2]. Studies have demonstrated that increasing numbers of ACEs are associated with a higher risk of depression and suicidal ideation [6,11]. Adults who have undergone more than six ACEs are at a higher risk of experiencing severe depressive symptoms [11]. Recent research has focused on the connection between mental health issues and ACEs among cancer survivors. Both child and adult cancer survivors who have experienced ACEs have reported elevated levels of physical and social impairments during and after cancer treatment [12,13].

Three childhood experiences that are part of ACEs—sexual abuse, physical abuse, and having family members with mental illnesses—are highly correlated with depression according to previous studies [14,15,16,17,18,19]. A history of both sexual and physical abuse is associated with an earlier onset and longer duration of mental health illnesses [19]. Among these, sexual abuse has the strongest association with depression [18]. Additionally, sexual abuse is linked to a higher likelihood of lifetime suicide attempts, inpatient admissions for depression, and more severe depression symptoms [16]. Furthermore, affective disorders like depression are more common in individuals with a parent suffering from depression or other affective illnesses [17]. Growing up in a family with members affected by mental health issues also increases the likelihood of maltreatment [17].

Prior research has explored the correlation between ACEs and depression among the general populace, yet there is a notable scarcity of studies examining this connection among cancer survivors. This study utilizes the 2020 Behavioral Risk Factor Surveillance System (BRFSS) database to explore the relationship between ACEs and self-reported depression diagnosis and mental distress. Our objective is to examine the association between the number of ACEs, sexual abuse, physical abuse, and mental illness in the family and depression and mental distress among cancer survivors aged ≥18 years in the United States.

## 2. Methods

### 2.1. Data Source and Study Sample

This study utilized a cross-sectional analysis of data from the 2022 Behavioral Risk Factor Surveillance System (BRFSS), the largest ongoing health survey in the world. [20] Conducted by state health departments under the supervision of the Centers for Disease Control and Prevention (CDC), the BRFSS collects responses from over 500,000 individuals. It is a state-based telephone survey targeting non-institutionalized adults across the United States and its territories. The BRFSS consists of three main components: core modules, which are sets of questions consistently administered to all states and territories to establish national estimates; optional modules, comprising CDC-developed questions that states can include based on their priorities; and state-added questions, which are customized items specific to each state. The survey methodology has been detailed in previous publications. This study used open access, de-identified data, and was exempt from institutional review board approval. In 2022, several states—including Arkansas, Florida, Iowa, Nevada, North Dakota, Oregon, South Dakota, Virginia, Arizona, Ohio, New Jersey, and Oklahoma—incorporated state-specific questions on adverse childhood experiences (ACEs). We followed the National Cancer Institute’s definition of a cancer survivor “someone who has been diagnosed with cancer and is still alive, from the time of diagnosis until the end of their life”. Thus, in our study, cancer survivors were any respondents who indicated they had ever been diagnosed with cancer. BRFSS collected these with questions that asked if respondents had ever been told they had skin cancer or melanoma or any other types of cancer. Respondents who selected yes were included in the study as cancer survivors. Respondents aged 18 years or older (n = 14,132) who had a history of cancer diagnosis were included in the study.

### 2.2. Measures

#### 2.2.1. Depression and Mental Distress

The primary outcome variable was self-reported history of depression, henceforth referred to as “depression”, determined by a respondent’s answer to the question “Has a doctor, nurse, or other health professional ever informed you that you had a depressive disorder (including depression, major depression, dysthymia, or minor depression)?” Response options were “yes” and “no”, with those answering “yes” categorized as having a history of depression. Mental distress (the secondary outcome variable) was evaluated with the question “Now thinking about your mental health, which includes stress, depression, and problems with emotions, for how many days during the past 30 days was your mental health not good?” Responses to this question were coded as “Yes” if subjects reported 6 or more days of “not good” mental health in the preceding month and as “No” otherwise. The cutoff was based on previous literature [5].

#### 2.2.2. Adverse Childhood Experiences

The primary exposure variable in this study was the number of ACEs, which was derived from the Behavioral Risk Factor Surveillance System (BRFSS) ACE items, whose content and scoring have been previously detailed [21]. Three forms of abuse (physical, emotional, and sexual) and five types of household challenges, such as having family members with substance misuse, incarceration, or mental illness; parental divorce; and witnessing intimate partner violence. A response indicating the presence of an ACE was coded as one, while the absence of an ACE was coded as zero. The individual responses were then summed to generate a total ACE score, ranging from 0 to 11. Categories of ACE scores were defined as follows: zero; one to two; and three or more, based on previous literature [22].

The secondary exposure variables were three ACEs—sexual abuse, physical abuse, and mental illnesses in the family—selected because they are highly correlated with depression according to previous studies [14,15,16,17,18,19]. Mental illness in the family was assessed with the question “Did you live with anyone who was depressed, mentally ill, or suicidal?” and classified as yes or no. Physical abuse was assessed with the question “Not including spanking, (before age 18), how often did a parent or adult in your home ever hit, beat, kick, or physically hurt you in any way?” and classified as no for never or yes for at least once. Sexual abuse was assessed with the question “How often did anyone at least 5 years older than you or an adult, force you to have sex?” and classified as no for never or yes for at least once.

#### 2.2.3. Covariates

The following covariates, selected based on previous research, were used in the study: age (18–64, ≥65), sex (female, male), race/ethnicity (Non-Hispanic White, Others), marital status (married/partnered, not married), education level (college graduate, some college, <high school/high school graduate), income level (≥USD 100,000, USD 59,000–USD 99,999, <USD 50,000, missing), body mass index (BMI) (normal, overweight, obese), smoking status (never, former, current smoker, missing), health status (excellent/very good, good, fair/poor) [23,24,25,26,27]. Age and race/ethnicity were combined due to low frequency in the 18–39 age group and non-Hispanic Blacks, Hispanics, and non-Hispanic Other categories (see Appendix A for a detailed breakdown). In addition, we conducted a sensitivity analysis using the three age categories and the four race/ethnicity categories, and the main findings were similar to when the categories were combined (Appendix A).

### 2.3. Statistical Analysis

All analyses accounted for the complex survey sample design of BRFSS, utilizing SAS, version 9.4 (SAS Institute Inc., Cary, NC, USA). Chi-square tests were conducted to compare depression and respondent characteristics. A weighted multivariable logistic regression model assessed the association between the number of ACEs and depression while adjusting for covariates (including age, gender, race/ethnicity, marital status, education level, income level, BMI, smoking status, and health status). Subsequently, three separate multivariable models, each focusing on a selected ACE, were employed to explore the association between selected ACEs (mental illness in the family, physical abuse, and sexual abuse) and depression, also adjusting for the same covariates mentioned above. Finally, a multinomial logistic regression was utilized to examine the association between the number of ACEs and our secondary outcome, mental distress. A significance level of 2-sided *p* = 0.05 was used for statistical significance.

## 3. Results

This study included 14132 adult cancer survivors (60.1% aged 65 and above, 57.1% female, 81.5% non-Hispanic White, 62.0% married/partnered, 32.6% college graduates, 29.1% normal BMI, and 47.7% never smokers). Respondents’ characteristics overall and stratified by depression are shown in Table 1. Approximately 22% of respondents reported experiencing ≥3 ACEs, and 27% had experienced 1–2 ACEs. The prevalence of depression was 21.8%, and mental distress was 15.4%. In the unadjusted analyses, respondents who were diagnosed with depression were more likely to be younger, be female, be unmarried, have less than a college degree, earn less than USD 50,000 per year, be obese, be current smokers, and have fair/poor health status (Table 1). A higher proportion of respondents with ≥3 ACEs (40.8%) had depression, followed by those with 1–2 ACEs (18.7%) and those with no ACEs (10.9%).

The association between the number of ACEs and depression, adjusting for covariates, is shown in Table 2. In the multivariable analysis, compared with cancer survivors who had experienced zero ACEs, those who had experienced three or more ACEs (aOR = 3.94; 95% CI, 3.04–5.10) or one to two ACEs (aOR = 1.85; 95% CI, 1.47–2.32) had a higher likelihood of reporting depression. Survivors had a higher likelihood of reporting depression if they were ≥65 (aOR = 1.76; 95% CI, 1.46–2.12) vs. 18–64, not married (aOR = 1.47; 95% CI, 1.22–1.76) vs. married, obese (aOR = 1.49; 95% CI, 1.17–1.89) vs. normal BMI, former (aOR = 1.23; 95% CI, 1.01–1.49) or current smokers (aOR = 1.95; 95% CI, 1.46–2.60) vs. never smokers, and had good (aOR = 1.35; 95% CI, 1.10–1.67) or fair/poor health status (aOR = 2.58; 95% CI, 2.07–3.24) vs. excellent health status. On the other hand, males (aOR = 0.55; 95% CI, 0.45–0.66) had a lower likelihood of reporting depression compared to females.

Table 3 presents the results from the selected ACEs and their associations with depression. Unsurprisingly, cancer survivors who lived with anyone who was depressed, mentally ill, or attempted suicide as children had a higher likelihood of reporting depression compared to those who did not live with anyone with those conditions (aOR = 3.60; 95% CI, 2.84–4.57; Model 1). Cancer survivors who experienced physical abuse at home by their parents or other adults had a higher likelihood of reporting depression compared to those who did not experience physical abuse (aOR = 2.76; 95% CI, 1.88–4.07; Model 2). Cancer survivors who experienced any sexual abuse as children had a higher likelihood of reporting depression compared to those who did not experience any sexual abuse (aOR = 1.86; 95% CI, 1.49–2.33; Model 3). Compared with cancer survivors who had experienced 0 ACEs, those who had experienced ≥3 (aOR = 0.67; 95% CI, 0.48–0.93) had a lower likelihood of reporting mental distress (Appendix A).

## 4. Discussion

This research study investigated the association between ACEs and depression and mental distress among individuals who have survived cancer. In our cohort, approximately 22% reported depression. Among those with three or more ACEs, the depression rate was 41%, compared to 19% among those with one to two ACEs and 11% among those with no ACEs. This linear correlation aligns with earlier studies that have explored how the number of ACEs experienced relates to the onset of depression in adulthood [28,29]. We found that compared to survivors who had experienced no ACE, those who had experienced any ACEs, especially those with at least three ACEs, had a higher likelihood of reporting depression [16,30]. This finding is in line with other studies that have shown severe depressive symptoms were associated with three or more ACEs compared to those with no prior history of ACEs [28]. Surprisingly, however, we found that cancer survivors who experienced at least three ACEs had a lower likelihood of reporting mental distress than those who did not experience any ACEs. One possible explanation for this counterintuitive finding is a prior study that highlighted a differential relationship between ACEs and mental distress based on gender [31]. Specifically, the study found that females had a higher likelihood of experiencing mental distress when they had lost both parents or faced sexual abuse, emotional violence, and interpersonal violence. In contrast, these ACEs were not associated with mental distress in males [31]. However, physical violence and community violence were linked to mental distress in both genders [31]. This suggests that the type of ACEs, some of which we did not have in our dataset, and the participant’s gender may have had a significant differential effect on mental distress. Our unexpected result might be attributed to the unique combination of ACEs we examined and the gender differences we did not explore, as this was not a focus of our hypothesis [31]. Future research should examine gender differences in relation to specific ACEs, rather than relying solely on aggregate measures. These findings highlight the impact that negative early life experiences have on mental health outcomes and may contribute to how cancer survivors cope with their diagnosis and treatment.

This study revealed that cancer survivors who resided with individuals experiencing depression, mental illness, or a history of childhood suicide attempts were more likely to report depression compared to those without such household members. These factors contribute to household dysfunction, which has been linked to elevated levels of suicidal thoughts and suicide attempts in affected individuals [2,32,33,34]. Previous research suggests that being exposed to trauma at an early age predisposes victims to develop feelings of learned helplessness which ultimately precedes psychopathologies including post-traumatic stress disorder (PTSD), depression, and anxiety. Other studies have suggested that ACEs, exposure to early violence, and other household dysfunction may alter the neurochemistry of the brain and possibly the parenchyma itself [34,35]. Specifically, volume changes or decreases in the brain parenchyma within key areas that are responsible for cognition, behavior, decision making, and mood may contribute to emotional and social deregulation as an adult and an inability to cope with stressors within the home [35]. Another aspect of household dysfunction recognized as an adverse childhood experience (ACE) is sexual abuse. In our study sample, cancer survivors who endured sexual abuse during childhood were more likely to report depression compared to those who did not experience such abuse. This finding is consistent with various childhood studies indicating that nonconsensual sexual contact or abuse increases the risk of mental health disorders in young adults [36,37,38]. Cancer survivors are a particularly vulnerable subgroup of people who have experienced ACEs and have the additional psychosocial burden of receiving a cancer diagnosis followed by enduring aggressive cancer treatments.

The existing literature extensively investigates major depressive disorder as the most prevalent mental health condition affecting both young individuals and adults [39]. It significantly contributes to disability-adjusted life years (DALY) more than any other mental illness and serves as a primary risk factor for suicidal ideation and attempts [6,40]. Recent studies indicate that approximately 20% of adolescents and young adults in America undergo depression [41]. This condition during childhood and adolescence detrimentally impacts the quality of life in adulthood and contributes to the development of adverse health outcomes [33,42,43]. The co-occurrence of depression with various chronic diseases such as heart disease, diabetes mellitus, stroke, and cancer is well documented in the literature, highlighting the profound influence of mental health on physical well-being and personal welfare [41]. Recent research investigating the development of cancer in individuals with ACEs suggests that subsequent alcohol and tobacco use create a chronic pro-inflammatory environment [44]. The findings in this study involving cancer survivors support previous research results regarding the association between ACEs and depression in adulthood [11,45]. Further studies are necessary to fully understand the impact of different types of ACEs on the mental and health outcomes of cancer survivors.

### Limitations

Several limitations should be considered when interpreting the findings of this study. Firstly, reliance on self-reported data introduces susceptibility to bias, such as potential misclassification of depression or recall bias regarding adverse childhood experiences (ACEs) and/or depression symptoms. Notably, there was a lack of information regarding the clinician responsible for diagnosing depression, and whether such diagnoses adhered to DSM-5 criteria, thereby allowing for the possibility of misdiagnosis. Furthermore, the self-reporting of depression may have led to both overestimation and underestimation of ACEs exposure and mental health challenges. Moreover, the study’s utilization of cross-sectional analysis prohibits the establishment of temporal significance or causality for observed outcomes. Regarding the timing of depression diagnoses, the survey data did not capture when participants were informed of their condition, whether in adolescence, young adulthood, or another period. The duration of depression following diagnosis remains unknown. Consequently, this study lacks the necessary data to ascertain if participants still experience depression in older adulthood, precluding any assertions regarding current depressive status. Also, because this study represents 12 states in the US, it may not be truly representative of the US population. Furthermore, there were instances of missing data on adverse childhood experiences (ACEs), although our analysis revealed no discernible difference in depression levels between respondents who reported ACEs and those who did not. Additionally, within the individual counts of ACEs, some may have stronger associations than others, and their distribution across ACE categories may not be uniform, considering count size.

## 5. Conclusions

In summary, cancer survivors who experienced adverse childhood experiences were more likely to report depression compared to those who had not experienced any ACE. On the other hand, survivors who experienced ≥3 ACEs were less likely to report mental distress compared to those who had not experienced any ACE. This research underscores the influence of ACEs on mental health among adult cancer survivors. By comprehending the subtleties of the connection between ACEs and mental health conditions in this group, it facilitates a deeper understanding of how ACEs affect depression in adulthood among cancer survivors. Furthermore, it may instigate further research aimed at enhancing existing practices targeted at addressing this issue.

## Figures and Tables

**Table 1 cancers-16-03290-t001:** Characteristics of survey respondents, overall and stratified by depression, 2022 Behavioral Risk Factor Surveillance System (n = 14,132).

	Frequency (Weighted Percent)			*p*-Value ^$^
	Overall	Depression		
		Yes	No	
**Depression**				
Yes	2762 (21.8)			
No	11,306 (78.2)			
**Mental distress**				
Yes	9702 (19.2)			
No	2128 (15.4)			
Missing	9708 (65.4)			
**# of ACEs**				<0.0001
Zero	4871 (28.3)	506 (10.9)	4349 (89.1)	
1–2	4169 (27.4)	775 (18.7)	3376 (81.3)	
≥3	2763 (22.2)	1059 (40.8)	1689 (59.2)	
Missing	2329 (22.1)	422 (20.3)	1892 (79.7)	
**Age at survey**				<0.0001
18–64	4202 (39.9)	1227 (29.8)	2956 (70.2)	
≥65	9712 (60.1)	1509 (16.7)	8159 (83.3)	
**Gender**				<0.0001
Female	7978 (57.1)	1901 (26.3)	6042 (73.7)	
Male	6154 (42.9)	861 (15.8)	5264 (84.2)	
**Race/ethnicity**				0.0013
Non-Hispanic White	12660 (81.5)	2385 (20.5)	10227 (79.5)	
Others	1472 (18.5)	377 (27.4)	1079 (72.6)	
**Marital status**				<0.0001
Married/partnered	7990 (62.0)	1252 (17.2)	6711 (82.8)	
Not married	6044 (38.0)	1490 (29.2)	4519 (70.8)	
**Education level**				0.0019
College graduate	5907 (32.6)	994 (18.4)	4896 (81.6)	
Some college	4116 (32.3)	895 (22.9)	3204 (77.1)	
<High school/HS graduate	4058 (35.1)	864 (24.0)	3168 (76.0)	
**Income level**				<0.0001
>=USD 100,000	2483 (20.0)	317 (14.5)	2162 (85.5)	
USD 50,000 to 99,999	3566 (24.1)	606 (20.4)	2948 (79.6)	
<USD 50,000	4946 (33.2)	1342 (29.0)	3579 (71.0)	
Missing	3137 (22.7)	497 (18.9)	2617 (81.1)	
**Body mass index**				<0.0001
Normal	4084 (29.1)	671 (19.3)	3396 (80.7)	
Overweight	4799 (33.3)	836 (19.3)	3943 (80.7)	
Obese	4123 (29.1)	1060 (27.9)	3050 (72.1)	
Missing	1126 (8.5)	195 (18.9)	917 (81.1)	
**Smoking status**				<0.0001
Never	6910 (47.7)	1133 (17.5)	5757 (82.5)	
Former	4940 (34.1)	973 (22.1)	1509 (77.9)	
Current	1361 (10.7)	491 (40.9)	859 (59.1)	
Missing	921 (7.5)	165 (20.3)	748 (79.7)	
**Health status**				<0.0001
Excellent/very good	5533 (39.8)	653 (14.3)	4863 (85.7)	
Good	4827 (33.2)	891 (20.3)	3916 (79.7)	
Fair/poor	3772 (27.0)	1218 (34.6)	2527 (65.4)	

^$^ *p*-value based on the Rao–Scott chi-square tests; ACE = adverse childhood experience.

**Table 2 cancers-16-03290-t002:** Multivariable logistic regression model estimating the association between adverse childhood experience and depression, 2022 Behavioral Risk Factor Surveillance System (n = 14,132).

	aOR (95% CI)
**# of ACEs**	
Zero	Reference
1–2	**1.85 (1.47, 2.32)**
≥3	**3.94 (3.04, 5.10)**
Missing	**2.10 (1.56, 2.82)**
**Age at survey**	
18–64	Reference
≥65	**1.76 (1.46, 2.12)**
**Gender**	
Female	Reference
Male	**0.55 (0.45, 0.66)**
**Race/ethnicity**	
NH-White	Reference
Other	1.04 (0.79, 1.35)
**Marital status**	
Married/partnered	Reference
Not married	**1.47 (1.22, 1.76)**
**Education level**	
College graduate	Reference
Some college	0.85 (0.69, 1.04)
<High school/HS graduate	0.73 (0.59, 0.91)
**Income level**	
>=USD 100,000	Reference
USD 50,000 to 99,999	**1.45 (1.11, 1.90)**
<USD 50,000	**1.56 (1.17, 2.09)**
Missing	1.29 (0.93, 1.78)
**Body mass index**	
Normal	Reference
Overweight	1.18 (0.94, 1.48)
Obese	**1.49 (1.17, 1.89)**
Missing	1.11 (0.73, 1.67)
**Smoking status**	
Never	Reference
Former	**1.23 (1.01, 1.49)**
Current	**1.95 (1.46, 2.60)**
Missing	1.04 (0.67, 1.62)
**Health status**	
Excellent/very good	Reference
Good	**1.35 (1.10, 1.67)**
Fair/poor	**2.58 (2.07, 3.24)**

**Table 3 cancers-16-03290-t003:** Multivariable logistic regression models estimating the association between three selected adverse childhood experiences and depression, 2022 Behavioral Risk Factor Surveillance System (n = 14,132).

	aOR (95% CI)		
	Model 1	Model 2	Model 3
**Mental illness**			
No	Reference		
Yes	3.60 (2.84, 4.57)		
Missing	1.36 (1.03, 1.76)		
**Physical abuse**			
No		Reference	
Yes		2.76 (1.88, 4.06)	
Missing		1.19 (0.93, 1.52)	
**Sexual abuse**			
No			Reference
Yes			1.86 (1.49, 2.33)
Missing			1.24 (0.96, 1.61)

All models adjusted for age at survey, gender, race/ethnicity, marital status, education level, income level, body mass index, smoking status, and health status.

## Data Availability

The data presented in this study are available in the Behavioral Risk Factor Surveillance System. These data were derived from the following resources available in the public domain: CDC—2022 BRFSS Survey Data and Documentation.

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
