# Peer review of "Adverse Childhood Events Significantly Impact Depression and Mental Distress in Adults with a History of Cancer"

_cancers, 2024, doi:10.3390/cancers16193290_

Round 1

Reviewer 1 Report

Comments and Suggestions for Authors

This paper focused on the relationship between Adverse Childhood Experiences (ACEs) including sexual abuse, physical abuse and familial mental illness, to depression and mental distress amongst adult cancer survivors. The introduction provided adequate background in a clear and concise manner. Methods utilized in this study, including data analytic methodologies were appropriate to the study aims and data reporting was clear. One of the results, however, seemed counterintuitive. Specifically, the finding that cancer survivors who had experiences three or more ACEs were less likely to report mental distress. While the authors mention that this was a surprise finding, they did not make an attempt at explaining this in the discussion section. This reader would have welcomed an attempt at an explanation. A strength of the discussion section is that the authors discuss their major finding in both psychological and neurobiological terms. The authors also mention that future studies should try to fully understand how different types of ACEs may impact both mental and health outcomes in cancer survivors. The limitations section seemed both honest and carefully thought-out. All in all, this was a well conceptualized study which yielded clinically useful data.

Author Response

Dear editor,

Thank you for the opportunity to revise our manuscript “Adverse childhood events significantly impact depression and mental distress in adults with a history of cancer.” We appreciate the careful review and constructive suggestions. We believe that the manuscript is much improved after making the further suggested edits. Following this letter are the comments from the reviewer/editors with our responses, including how and where the text was modified.

Thank you for your consideration.

Sincerely,

Oluwole A. Babatunde

Reviewer Comment

Author Response

Reviewer 1

This paper focused on the relationship between Adverse Childhood Experiences (ACEs) including sexual abuse, physical abuse and familial mental illness, to depression and mental distress amongst adult cancer survivors. The introduction provided adequate background in a clear and concise manner. Methods utilized in this study, including data analytic methodologies, were appropriate to the study aims and data reporting was clear. A strength of the discussion section is that the authors discuss their major findings in both psychological and neurobiological terms. The authors also mention that future studies should try to fully understand how different types of ACEs may impact both mental and health outcomes in cancer survivors. The limitations section seemed both honest and carefully thought-out. All in all, this was a well conceptualized study which yielded clinically useful data.

Thank you, we appreciate your feedback.

One of the results, however, seemed counterintuitive. Specifically, the finding that cancer survivors who had experiences three or more ACEs were less likely to report mental distress. While the authors mention that this was a surprise finding, they did not make an attempt at explaining this in the discussion section. This reader would have welcomed an attempt at an explanation.

We added the following paragraph:

One possible explanation for this counterintuitive finding is a prior study that highlighted a differential relationship between ACEs and mental distress based on gender. [1] Specifically, the study found that females had a higher likelihood of experiencing mental distress when they had lost both parents or faced sexual abuse, emotional violence, and interpersonal violence. In contrast, these ACEs were not associated with mental distress in males. [1] However, physical violence and community violence were linked to mental distress in both genders. [1]This suggests that the type of ACEs, some of which we did not have in our dataset, and the participant's gender may have had a significant differential effect on mental distress. Our unexpected result might be attributed to the unique combination of ACEs we examined and the gender differences we did not explore, as this was not a focus of our hypothesis. [1] Future research should examine gender differences in relation to specific ACEs, rather than relying solely on aggregate measures.

Reviewer 2

1- (160). In the methodology section there is no description of the sample and although the data appear later in Table 1, it would be better to present the sample separately.

We appreciate your comments and have addressed them point by point below. The characteristics of the study sample are described in the first paragraph of the results as is usually the case. However, we have added a couple sentences on how we identified cancer survivors and the study sample in the methods section.

“We followed the National Cancer Institute’s definition of a cancer survivor “someone who has been diagnosed with cancer and is still alive, from the time of diagnosis until the end of their life”. Thus, in our study, cancer survivors were any respondent which indicated they had ever been diagnosed with cancer. BRFSS collected these with questions that asked if respondents had ever been told they had skin cancer or melanoma or any other types of cancer. Respondents who selected yes were included in the study as cancer survivors.”

2- (168). There is a very high percentage of missing responses to the Mental distress variable. A precise explanation of this data should be given.

We do not know the reason why there is a high percentage of missingness for the mental distress variable. BRFSS did not provide any reason, the question was asked to all participants without any skip logic. Thus, we can only assume that several respondents were perhaps not comfortable answering the question.

3- (201). The established age range 18-65 is too wide. There should be at least three ranges representing: youth, adulthood and old age. This would give greater richness to the analysis and would make it possible to know if there are differences in impact with the passage of time.

Thank you for the We had three age groups (18-39, 40-64; >=65) but the 18-39 age group had a small sample (only 4% (n=352), see tables at the end of response document). To make sure the analysis is more robust without wide confidence intervals, we combined 18-39 and 40-64. We conducted a sensitivity analysis using the three age categories and the results were similar to when they were combined. We have added more information about the combination in the methods section. The sensitivity results are now presented in the supplemental material.

4- (201). The same occurs with the variable race NH-White vs. Others is excessive in an American sample at least four different races should appear: whites, blacks, Asians and Latinos.

Like age, BRFSS categorized race/ethnicity into 4 groups (NH-White, NH-Black, Hispanic, NH-Other) but the other groups other than NH-Whites were small (see tables at the end of response document). To make sure the analysis is more robust without wide confidence intervals, we combined all the other race/ethnicities. We conducted a sensitivity analysis using the three age categories and the results were similar to when they were combined. We have added more information about the combination in the methods section. The sensitivity results are now presented in the supplemental material.

Reviewer 3

If the respondents are from the 12 states listed on lines 120-122, then the sample is not truly representative of the US population.  Large and important states, most notably, California, NY and Texas are missing.

We have read through the discussion and did not mention explicitly that this is a representative study of US sample. We also added the sentence below to limitations.

Also, because this study represents 12 states in the US, it may not be truly representative of the US population.

The authors based their cut-off points (lines 137 and 150) on past research, which is fine; however, a justification in the present paper must still be provided for these cut-off points.  Readers should not have to carry the burden of searching the literature for the authors' methodological decisions.

This previous study found that experiencing ≥3 ACEs was associated with 145% increased odds of reporting at least one health-risk behavior (OR = 2.45, 95% CI [1.78-3.38]) when compared to those without a history of ACEs. [2]

There is a typo on line 139: “if” should be “of”

Fixed.

The authors should tell us more about the target population (line 183).  What is the definition of “cancer survivor”?  The category of "cancer survivor" seems very broad.  It could include, at one extreme, those treated with one surgery for one small basal cell skin cancer to, at another extreme, those treated with surgery, radiation, and chemo for stage 4 breast cancer.  What experience qualifies one to be called a "cancer survivor"?

We have added a couple sentences on how we defined cancer survivors in the methods section. We followed the National Cancer Institute’s definition of a cancer survivor “someone who has been diagnosed with cancer and is still alive, from the time of diagnosis until the end of their life”. Thus, in our study, cancer survivors were any respondent which indicated they had ever been diagnosed with cancer.

“We followed the National Cancer Institute’s definition of a cancer survivor “someone who has been diagnosed with cancer and is still alive, from the time of diagnosis until the end of their life”. Thus, in our study, cancer survivors were any respondent which indicated they had ever been diagnosed with cancer. BRFSS collected these with questions that asked if respondents had ever been told they had skin cancer or melanoma or any other types of cancer. Respondents who selected yes were included in the study as cancer survivors.”

The counterintuitive finding of lower odds of mental distress for those reporting three or more ACE (lines 226-228) must be explained in detail (see also lines 244-246).

We added the following paragraph:

One possible explanation for this counterintuitive finding is a prior study that highlighted a differential relationship between ACEs and mental distress based on gender. [1] Specifically, the study found that females had a higher likelihood of experiencing mental distress when they had lost both parents or faced sexual abuse, emotional violence, and interpersonal violence. In contrast, these ACEs were not associated with mental distress in males. [1] However, physical violence and community violence were linked to mental distress in both genders. [1]This suggests that the type of ACEs, some of which we did not have in our dataset, and the participant's gender may have had a significant differential effect on mental distress. Our unexpected result might be attributed to the unique combination of ACEs we examined and the gender differences we did not explore, as this was not a focus of our hypothesis. [1] Future research should examine gender differences in relation to specific ACEs, rather than relying solely on aggregate measures.

The biggest problem I have with the paper is that it does not address the issue of “vulnerability” of cancer survivors (lines 267-269) because the analysis does not compare cancer survivors with those who have never experienced cancers.  The key question should be:  all else being equal, does the experience of cancer treatment and survivorship make one more vulnerable to depression/mental distress?  That question cannot be addressed by the study’s research design.  A comparative analysis is needed.

We understand your concern. Several studies have been done comparing mental health outcomes (depression, distress, anxiety etc) between cancer survivors and individuals without a cancer diagnosis. These studies report that cancer survivors have higher mental health problems than the general population. Therefore, the goal of this paper was to examine the effect of childhood ACE experiences and depression among cancer survivors. Similarly, several studies have looked at the association between ACEs and depression among the general population without cancer. Thus, our study was to fill out the gap of examining association between ACE and depression among cancer survivors. We hope this study contributes useful information to the body of literature.

Editorial Comment

Delete running title

The running title is deleted.

Rewrite highlighted text

We have rewritten the highlighted portions.

Fixed references

We have fixed the references.

Supplemental Table 1. Granular age and race/ethnicity distribution.

N (%)

Age at time of survey

18-39

352 (5.4)

40-64

3850 (34.5)

≥65

9712 (60.1)

Race/ethnicity

Non-Hispanic White

12660 (81.5)

Non-Hispanic Black

349 (4.5)

Hispanic

330 (6.6)

Non-Hispanic Other

793 (7.4)

Supplemental Table 2. Multivariable logistic regression model estimating association between adverse childhood experience and depression using granular age and race/ethnicity.

aOR (95% CI)

# of ACEs

Zero

1-2

≥3

Missing

Reference

1.84 (1.47, 2.31)

3.73 (2.89, 4.81)

2.08 (1.55, 2.80)

Age at survey

18-39

2.87 (1.80, 4.56)

40-64

1.68 (1.40, 2.02)

≥65

Reference

Race/ethnicity

Non-Hispanic White

Reference

Non-Hispanic Black

0.74 (0.43, 1.25)

Hispanic

0.73 (0.47, 1.14)

Non-Hispanic Other

1.68 (1.17, 2.42)

  1. Perry Mohling, E.W., et al., Adverse childhood experiences, mental distress, self-harm and suicidality, and cumulative HIV risk by sex in Lesotho. Child Abuse Negl, 2024. 150: p. 106701.
  2. Sarkar, S., et al., Association between adverse childhood experiences and self-reported health-risk behaviors among cancer survivors: A population-based study. PLoS One, 2024. 19(3): p. e0299918.

Reviewer 2 Report

Comments and Suggestions for Authors

1- (160). In the methodology section there is no description of the sample and although the data appear later in Table 1, it would be better to present the sample separately.

2- (168). There is a very high percentage of missing responses to the Mental distress variable. A precise explanation of this data should be given.

3- (201). The established age range 18-65 is too wide. There should be at least three ranges representing: youth, adulthood and old age. This would give greater richness to the analysis and would make it possible to know if there are differences in impact with the passage of time.

4- (201). The same occurs with the variable race NH-White vs. Others is excessive in an American sample at least four different races should appear: whites, blacks, Asians and Latinos.

Author Response

(The authors gave the same response as above.)

Author Response

(The authors gave the same response as above.)

Reviewer 3 Report

Comments and Suggestions for Authors

The paper’s topic is interesting, but I have some questions and concerns.

If the respondents are from the 12 states listed on lines 120-122, then the sample is not truly representative of the US population.  Large and important states, most notably, California, NY and Texas are missing.

The authors based their cut-off points (lines 137 and 150) on past research, which is fine; however, a justification in the present paper must still be provided for these cut-off points.  Readers should not have to carry the burden of searching the literature for the authors' methodological decisions.

There is a typo on line 139: “if” should be “of”

The authors should tell us more about the target population (line 183).  What is the definition of “cancer survivor”?  The category of "cancer survivor" seems very broad.  It could include, at one extreme, those treated with one surgery for one small basal cell skin cancer to, at another extreme, those treated with surgery, radiation, and chemo for stage 4 breast cancer.  What experience qualifies one to be called a "cancer survivor"?

The counterintuitive finding of lower odds of mental distress for those reporting three or more ACE (lines 226-228) must be explained in detail (see also lines 244-246).

The biggest problem I have with the paper is that it does not address the issue of “vulnerability” of cancer survivors (lines 267-269) because the analysis does not compare cancer survivors with those who have never experienced cancers.  The key question should be:  all else being equal, does the experience of cancer treatment and survivorship make one more vulnerable to depression/mental distress?  That question cannot be addressed by the study’s research design.  A comparative analysis is needed.

Comments on the Quality of English Language

No problems

Author Response

(The authors gave the same response as above.)

Author Response

(The authors gave the same response as above.)

Round 2

Reviewer 3 Report

Comments and Suggestions for Authors

I still think there are too many problems with this paper, despite the authors' efforts to revise.

Comments on the Quality of English Language

No problems.